# REASONING UNDER UNCERTAINTY: EXPLORING PROBABILISTIC REASONING CAPABILITIES OF LLMS

## ABSTRACT

Despite widespread success in language understanding and generation, large language models (LLMs) exhibit unclear and often inconsistent behavior when faced with tasks that require probabilistic reasoning. In this work, we present the first comprehensive study of the reasoning capabilities of LLMs over explicit discrete probability distributions. Given observations from a probability distribution, we evaluate models on three carefully designed tasks—mode identification, maximum likelihood estimation, and sample generation—by prompting them to provide responses to queries about either the joint distribution or its conditionals. These tasks thus probe a range of probabilistic skills, including frequency analysis, marginalization, and generative behavior. Through comprehensive empirical evaluations, we demonstrate that there exists a clear performance gap between smaller and larger models, with the latter demonstrating stronger inference and surprising capabilities in sample generation. Furthermore, our investigations reveal notable limitations, including sensitivity to variations in the notation utilized to represent probabilistic outcomes and performance degradation of over 60% as context length increases. Together, our results provide a detailed understanding of the probabilistic reasoning abilities of LLMs and identify key directions for future improvement.

## 1 INTRODUCTION

Large Language Models (LLMs) have achieved impressive performance across a wide range of Natural Language Processing (NLP) tasks, including question answering, summarization, and language understanding (Devlin et al., 2019; Raffel et al., 2020; Chowdhery et al., 2023; Joshi et al., 2017). Despite these advances, LLMs continue to show notable limitations in probabilistic reasoning, as highlighted by several studies (Freedman & Toni, 2025; Gu et al., 2024; Ball et al., 2024). Existing research has largely examined LLMs' abilities in logical reasoning, numerical problem-solving, and Bayesian inference (Chen et al., 2022; Ozturkler et al., 2022; Nafar et al., 2023; 2025; Qiu et al., 2025), but their ability to understand probability distributions has received far less attention (Gu et al., 2024; Paruchuri et al., 2024). Exploring this capability is crucial, as reasoning over distributions lies at the core of many tasks involving uncertainty, prediction, and decision-making (Jia et al., 2024; Liu et al., 2024a; Schrader et al., 2024). This gap leads to an important question: *when provided with samples drawn from a distribution, can an LLM identify its structure, estimate its parameters, and produce responses consistent with that distribution?*

To address this, we develop a structured framework that prompts models with observations drawn from the joint probability distribution of discrete random variables and evaluates their responses to queries about joint and conditional distributions. Our evaluation suite consists of three complementary tasks, mode identification, maximum likelihood estimation, and sample generation. Together, these tasks probe distinct but interconnected skills: recognizing the most likely outcomes, estimating underlying probabilities, and producing samples that align with the target distribution. By covering both joint and conditional settings, our study provides a broad view of how well LLMs can approximate key features of probability distributions without requiring finetuning on probabilistic datasets.

Our experiments reveal several important trends. Larger models, and those distilled from them, significantly outperform smaller ones in most settings. Across all three tasks, models perform better on joint distributions than on conditional ones, which demand deeper reasoning and additional computation. Some models show surprising strengths in sampling, generating outcomes that closely

match the expected distribution. On the other hand, we identify persistent weaknesses. Performance is highly sensitive to how outcomes of the distributions are represented in the prompt, suggesting that even superficial variations in notation can significantly influence reasoning. Counting also remains a fundamental challenge: as the number of samples in the context grows, models lose track of frequencies, leading to inaccurate estimates. Conditional reasoning is particularly challenging in zero-shot settings, but providing a simple in-context example improves the performance significantly. More broadly, while LLMs can often handle probabilistic tasks using their background knowledge, they struggle as the tasks become more complex. In these cases, they need additional support, whether through external tools like a code interpreter or in-context examples, to achieve reliable results.

In summary, this work makes the following contributions:

- We present the first large-scale evaluation of LLMs' ability to understand discrete probability distributions across a variety of settings.
- We introduce a suite of complementary tasks—mode identification, maximum likelihood estimation, and sampling—that capture distinct aspects of probabilistic reasoning.
- We systematically compare a broad range of LLMs, revealing substantial performance gaps across model sizes and unexpected strengths in sampling.
- We highlight key limitations, including sensitivity to notation of probabilistic outcomes, difficulties with counting, and challenges with long-context reasoning.

## 2 RELATED WORK

**In-Context Learning:** After the discovery of in-context learning (ICL) capabilities in large language models (LLMs), there has been a surge of interest in enhancing inference in zero-shot and few-shot settings. This paradigm enables models to perform tasks by conditioning on input examples without explicit parameter updates. Foundational studies have explored the mechanisms and effectiveness of ICL in various contexts (Brown et al., 2020; Min et al., 2022; Xie et al., 2021; Dai et al., 2022; Kojima et al., 2022; Wang et al., 2022). Building upon ICL, follow-up work has introduced methods such as Chain-of-Thought (CoT) and Tree-of-Thoughts (ToT) (Yao et al., 2023; Wei et al., 2022), which aim to extend LLMs' reasoning capabilities and improve performance on more complex problems.

**Probabilistic Reasoning:** Despite these advances, probabilistic reasoning and handling uncertainty remain challenging for LLMs (Nafar et al., 2023; Kadavath et al., 2022). Recent evaluations (Freedman & Toni, 2025) found that current models frequently violate basic probability rules such as complementarity and monotonicity, and (Ozturkler et al., 2022) proposed a structured approach, ThinkSum, decomposing probabilistic reasoning into distinct retrieval and aggregation stages. Other studies show that while LLMs understand probability concepts, they struggle to generate samples aligning with specified distributions (Gu et al., 2024). Bayesian reasoning has also emerged as a promising direction. The BLInD dataset was introduced to evaluate LLMs' ability to perform Bayesian inference (Nafar et al., 2025), and training LLMs to imitate an optimal Bayesian model has been shown to improve their probabilistic reasoning capabilities (Qiu et al., 2025).

**Numerical Reasoning:** Another important area of study is LLMs' quantitative reasoning. Several works have shown that numerical reasoning skills can be effectively enhanced through automated data generation and targeted training (Geva et al., 2020; Lewkowycz et al., 2022). The Program of Thoughts (PoT) approach expresses numerical reasoning as executable programs, enabling models to solve complex calculations (Chen et al., 2022). While some studies have shown that large models can reason about continuous distributions when properly guided (Paruchuri et al., 2024), others highlight that LLMs struggle to infer probabilities from raw data or tables without special training (Liu et al., 2024b), and their probability estimates can be biased or unreliable (Wang et al., 2025). Counting tasks also proved to be challenging (Fu et al., 2024; Ball et al., 2024), with difficulties partly attributed to architectural constraints and tokenization issues inherent to transformers (Zhang et al., 2024; Vaswani et al., 2017).

In summary, while LLMs have shown promising capabilities in probabilistic reasoning and Bayesian inference, their ability to understand discrete probability distributions, especially joint and conditional relationships, remains largely unexplored. This gap motivates our work, where we provide a focused and extensive study of LLMs' understanding of discrete joint and conditional probability distributions.

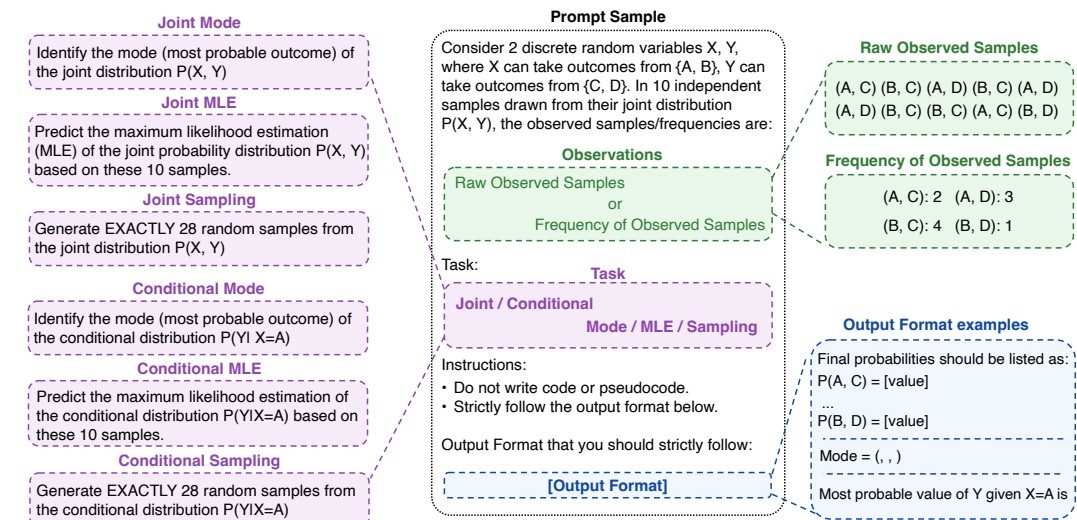

Figure 1: Prompt structure for evaluating LLMs on probabilistic reasoning tasks: each prompt includes definitions of the random variables, observed samples or frequencies, a task specification (e.g., mode, MLE, sample generation), and output formatting instructions.

# 3 PROBABILISTIC REASONING FRAMEWORK

In this section, we present our framework for evaluating LLMs' capability to understand probability distributions. We begin with the mathematical preliminaries and describe the prompt format used in our experiments. Then, we formalize the three structured tasks that form the core of our evaluation, and finally present our comprehensive evaluation pipeline. In this work, we decided to focus on discrete probability distributions, as methods and evaluation strategies for continuous distributions differ enough that addressing both types in one paper would risk superficial coverage.

## 3.1 PRELIMINARIES ON DISCRETE PROBABILITY DISTRIBUTIONS

We formalize the probabilistic setup used in our study, wherein we assess whether LLMs can understand discrete probability distributions by evaluating their performance across several tasks. Let $X = (X_1, X_2, \ldots, X_N)$ denote a tuple of $N$ categorical random variables, where each $X_i$ takes values in a finite label set $\mathcal{L}_i$. We refer to the elements of $\mathcal{L}_i$ as the possible *labels* of $X_i$. An outcome of the joint random variable $X$ is denoted by $\mathbf{x} = (x_1, x_2, \ldots, x_N)$, where $x_i \in \mathcal{L}_i$ for all $i \in [N]$. Let $\mathcal{L} = \mathcal{L}_1 \times \mathcal{L}_2 \times \cdots \times \mathcal{L}_N$ denote the set of all possible joint outcomes. The size of the joint outcome space is then given by

$$|\mathcal{L}| = \prod_{i=1}^{N} |\mathcal{L}_i|.$$

To grant the LLM access to a distribution, we provide a set of $K$ independent observations drawn from the joint distribution. Let $\mathcal{O} = \{\mathbf{x}^{(1)}, \mathbf{x}^{(2)}, \ldots, \mathbf{x}^{(K)}\}$ denote the observed samples. This information is incorporated into the LLM's context using one of two following formats: (1) by providing the empirical frequency of each possible outcome $\mathbf{x}$ in $\mathcal{O}$, or (2) by directly including the set of observations $\mathcal{O}$ in the prompt.

## 3.2 PROMPT FORMAT

As shown in Figure 1, each prompt begins with a fixed segment that defines the random variables $X_1, X_2, \ldots, X_N$, their corresponding label sets $\mathcal{L}_1, \mathcal{L}_2, \ldots, \mathcal{L}_N$ and the joint probability distribution $P(X_1, X_2, \ldots, X_N)$. This is followed by observations from the joint distribution, provided either as empirical frequencies or raw samples. The prompt concludes with a task-specific query and clear

instructions, explicitly discouraging code generation and specifying the expected output format. A complete list of prompts used for each task and setting is provided in Appendix B.

### 3.3 TASK DEFINITION AND PERFORMANCE METRICS

In this section, we describe our three structured tasks and the metrics used to assess LLMs' performance on them. We consider two settings: one focused on queries about joint distributions and the other on queries about conditional distributions.

**Mode Identification:** Given a set of $K$ observations $\mathcal{O} = \{\mathbf{x}^{(1)}, \ldots, \mathbf{x}^{(K)}\}$ provided to the model, this task evaluates whether the model can identify the mode of the empirical distribution.

*Joint task.* We measure if the model can find the mode of the joint distribution i.e.,

$$\hat{\mathbf{x}}_{\text{mode}} = \arg\max_{\mathbf{x} \in \mathcal{L}} \frac{1}{K} \sum_{k=1}^{K} \mathbb{1}[\mathbf{x}^{(k)} = \mathbf{x}],$$

*Conditional task.* We condition on a random variable $X_c$ and evaluate the mode of a query variable $X_q$. Given a conditioning value $X_c = x_c$ for some $x_c \in \mathcal{L}_c$, we define:

$$\hat{x}_q = \arg\max_{x_q \in \mathcal{L}_q} \hat{p}(X_q = x_q \mid X_c = x_c),$$

where $\hat{p}$ is the empirical conditional distribution estimated from the subset of $\mathcal{O}$ satisfying $X_c = x_c$. We report *accuracy* for this task, indicating whether the model correctly identifies the most likely outcome.

**Maximum Likelihood Estimation (MLE):** Given the observed samples $\mathcal{O}$, the model is asked to estimate the empirical distribution over possible outcomes, either for the joint distribution or in a conditional manner.

*Joint task:* The model estimates $\hat{p}(X = \mathbf{x})$ for all $\mathbf{x} \in \mathcal{L}$. We measure the total variation distance (TVD) between the predicted empirical distribution $\hat{p}$ and the true empirical distribution $p$ from $\mathcal{O}$, i.e.,

$$\frac{1}{2} \sum_{\mathbf{x} \in \mathcal{L}} \left| p(X = \mathbf{x}) - \hat{p}(X = \mathbf{x}) \right|.$$

*Conditional task:* Given $X_c = x_c$, the model estimates $\hat{p}(X_q = x_q \mid X_c = x_c)$ for all $x_q \in \mathcal{L}_q$. We once again report the total variation distance for this task, i.e.,

$$\frac{1}{2} \sum_{x_q \in \mathcal{L}_q} \left| p(X_q = x_q \mid X_c = x_c) - \hat{p}(X_q = x_q \mid X_c = x_c) \right|.$$

**Sample Generation:** In this task, the model is prompted to generate new samples from the joint or conditional distribution based on the provided observations $\mathcal{O}$. Unlike the previous tasks focused on estimation or prediction, this task evaluates the model's generative capability, i.e., whether it can reproduce the empirical distribution through sampling.

*Joint task:* The model is prompted to generate $K'$ samples based on the joint distribution provided in the context. Let $\hat{\mathcal{O}}$ be the samples generated by the LLM, and let $q$ denote their empirical frequency distribution. We report the total variation distance, which is defined in this setting as:

$$\frac{1}{2} \sum_{\mathbf{x} \in \mathcal{L}} \left| p(X = \mathbf{x}) - q(X = \mathbf{x}) \right|.$$

*Conditional task:* The model is prompted to generate $K'$ samples of the random variable $X_q$ conditioned upon $X_c = x_c$. Let $q(X_q \mid X_c = x_c)$ be the empirical frequency distribution over the samples generated by the LLM. We report the total variation distance, which is defined in this setting as:

$$\frac{1}{2} \sum_{x_q \in \mathcal{L}_q} \left| p(X_q = x_q \mid X_c = x_c) - q(X_q = x_q \mid X_c = x_c) \right|$$

By evaluating models on joint and conditional distributions, we examine their ability to handle probabilistic relationships in various settings. Each task focuses on a specific aspect, providing a comprehensive assessment of how well LLMs understand and process discrete probability distributions.

## 3.4 EVALUATION SETUP

As outlined in Section 3.2, output formats are explicitly defined in the prompts to guide models toward consistent responses. Through extensive prompt engineering, task-specific formats are identified that models generally find easier to follow. When responses match the expected format, final answers are extracted using regular expressions, and the corresponding evaluation metrics are applied as mentioned above. While larger models typically follow the instructions closely, smaller models often return answers in inconsistent or alternative formats. To handle such cases, a capable model, `Llama3.3-70B`, is used as an LLM judge. It receives the task description, the expected answer, and the model's response, and determines whether the output reflects the correct answer. This approach supports a robust evaluation and helps ensure the reliability of reported results. We provide additional details regarding the usage of the LLM Judge in Appendix D due to paucity of space.

## 4 ROBUSTNESS ANALYSIS

In this section, we examine pertinent factors that influence LLMs' performance on probabilistic reasoning tasks, focusing on three key aspects: (1) how performance changes with increasing task difficulty, (2) how sensitive models are to variations in the notation used to represent random variable outcomes, and (3) how models behavior differs when prompts provide raw sample observations rather than empirical frequencies. Together, these aspects highlight distinct challenges for LLMs and provide deeper insight into their strengths and limitations.

### 4.1 SCALING COMPLEXITY

To examine how task complexity affects model performance, we vary the size of the outcome or label space $|\mathcal{L}|$ for the joint distributions considered, which directly influences the difficulty of all three tasks. As shown in equation 3.1, increasing either the number of random variables $N$ or the number of labels $|\mathcal{L}_i|$ for each variable causes exponential or multiplicative growth of the joint space respectively, requiring models to reason over a larger set of outcomes and their associated probabilities. In the mode identification task, this expansion increases the number of candidates the model must compare, making it harder to identify the most probable outcome. In the MLE task, a larger joint space implies that the model must estimate a greater number of probabilities, making accurate prediction more challenging. Similarly, in the sampling task, the model is expected to generate samples that reflect a more complex distributed set of outcomes, demanding stronger generalization and a better understanding of the underlying probabilities.

### 4.2 LABEL SENSITIVITY

Our experiments reveal that LLMs are highly sensitive to the choice of labels utilized in the set $\mathcal{L}_i$, with performance varying significantly across different label categories. Interestingly, even when the set of labels and frequency counts are fixed, simply changing which outcome is associated with which frequency can lead to noticeable shifts in model performance. This behavior likely stems from biases in the training data and the tendency of the model to interpret labels as part of the linguistic context, rather than discrete outcomes in a probabilistic space. Additionally, sensitivity to label variation differs across models, with smaller models generally showing greater variability. To quantify this effect, we design a set of experiments to measure models' robustness to superficial changes in label notation.

### 4.3 ESTIMATIONS USING SAMPLES-IN-CONTEXT INSTEAD OF FREQUENCIES

In this setting, we explore the effect of including the raw observations $\{\mathbf{x}^{(1)}, \mathbf{x}^{(2)}, \ldots, \mathbf{x}^{(K)}\}$ in the input prompts, rather than their empirical frequencies. By doing so, models should count the frequencies of each outcome and solve the probabilistic tasks accordingly. This change shifts the problem from simple frequency analysis to counting and subsequent processing of the counts so obtained. The longer the context, the more challenging it becomes for the model to correctly interpret the frequency of occurrences, leading to a sharp decline in performance. In the next section, we illustrate a comparison between these two approaches and highlight the weaknesses of powerful models in counting and managing a large number of samples.

Table 1: Accuracy comparison on mode identification tasks across different joint sizes $|\mathcal{L}|$. *cond-mode* requires reasoning over conditional distributions, making it generally more challenging. While larger models or those distilled from them maintain robust performance across all settings, smaller models show a clear decline in accuracy as $|\mathcal{L}|$ increases.

| Model | joint-mode | | | | cond-mode | | | |
|---|---|---|---|---|---|---|---|---|
| | $\|\mathcal{L}\| = 12$ | $\|\mathcal{L}\| = 27$ | $\|\mathcal{L}\| = 36$ | $\|\mathcal{L}\| = 54$ | $\|\mathcal{L}\| = 12$ | $\|\mathcal{L}\| = 27$ | $\|\mathcal{L}\| = 36$ | $\|\mathcal{L}\| = 54$ |
| `Llama3.1-8B` | 0.59 | 0.57 | 0.35 | 0.21 | 0.83 | 0.79 | 0.68 | 0.62 |
| `Qwen2.5-7B` | 0.93 | 0.83 | 0.81 | 0.65 | 0.69 | 0.67 | 0.63 | 0.60 |
| `DeepSeek-R1-Distill-Qwen-7B` | 0.95 | 0.93 | 0.91 | 0.86 | 0.93 | 0.93 | 0.91 | 0.83 |
| `Llama3.3-70B` | 1.00 | 1.00 | 1.00 | 0.97 | 1.00 | 0.99 | 0.96 | 0.95 |
| `GPT-4o-mini` | 0.99 | 0.98 | 0.98 | 0.80 | 0.99 | 0.96 | 0.95 | 0.88 |
| `GPT-4.1-mini` | 1.00 | 1.00 | 0.99 | 0.96 | 1.00 | 1.00 | 1.00 | 1.00 |

## 5 EXPERIMENTS

This section presents the experimental setup for evaluating LLMs on probabilistic reasoning tasks. We introduce the models evaluated, describe key implementation details, and report results for the three core tasks. We then present additional analyses that explore factors influencing model performance.

### 5.1 MODEL SELECTION

In this study, we evaluate several instruction-finetuned LLMs, selected to represent a wide range of model sizes and capabilities. The models evaluated include `Llama3.1-8B`, `Llama3.3-70B` (Grattafiori et al., 2024), `Qwen2.5-7B` (Yang et al., 2024), `DeepSeek-R1-Distill-Qwen-7B` (Guo et al., 2025), `GPT-4o-mini` (Hurst et al., 2024), and `GPT-4.1-mini` (OpenAI, 2025). These models span a broad spectrum of parameter sizes, from 7B to 70B, offering a comprehensive comparison of how different architectures handle various settings.

### 5.2 EXPERIMENTAL SETUP

In our experiments, we follow the standard naming convention for random variables, using names such as $X, Y, Z$ and use English letters for the outcomes of the random variables . We set $K := 5 \times |\mathcal{L}|$, meaning the number of observations is five times larger than the joint space. We then select $|\mathcal{L}|$ values that sum to $K$, as frequencies, and randomly assign them to different joint outcomes. However, as discussed in Section 4.2, models are sensitive to how these frequencies are linked to specific outcomes, even when both the set of labels and frequency values are unchanged. To reduce the impact of this sensitivity, we create ten different ways of assigning frequencies to joint outcomes and generate ten prompts for each assignment, resulting in 100 prompts per task. This helps ensure that variations in outcome-frequency mapping minimally impacts the evaluation results. For larger $|\mathcal{L}|$, we carefully select the frequency values to generate a distribution with entropy comparable to that of the simpler distributions, ensuring consistency across experiments. Given the models' constraints in handling raw samples in the input, we incorporate the empirical frequency of observed samples in most experiments and analyze the comparison between the two approaches in Section 5.4.2.

### 5.3 TASK-SPECIFIC EXPERIMENTAL RESULTS

This section presents a detailed analysis of the experimental results for each task. To evaluate performance on conditional queries, one random variable $X_c$ and one of its possible outcomes $x_c$ are randomly selected as evidence. Another variable is then chosen as the query variable $X_q$ and the model is asked to solve one of the tasks based on the conditional distribution $P(X_q \mid X_c = x_c)$. For all tasks, each model is prompted 100 times using 10 different prompts, and the outputs are parsed to extract the final answer for each query. The details of different distributions can be found in Appendix C.

**Mode Identification:** In this task, models are asked to identify the mode of either the joint distribution *(joint-mode)* or a conditional distribution *(cond-mode)*. The *joint-mode* task requires selecting the outcome with the highest overall frequency, without additional computation. In contrast, the *cond-*

Table 2: Average TVD across 100 prompts for maximum likelihood estimation tasks at varying joint sizes $|\mathcal{L}|$. *joint-MLE* is consistently easier with near-zero TVD, while *cond-MLE* seems more challenging and highlights a clear gap between models.

| Model | joint-MLE | | | | cond-MLE | | | |
|---|---|---|---|---|---|---|---|---|
| | $|\mathcal{L}| = 12$ | $|\mathcal{L}| = 27$ | $|\mathcal{L}| = 36$ | $|\mathcal{L}| = 54$ | $|\mathcal{L}| = 12$ | $|\mathcal{L}| = 27$ | $|\mathcal{L}| = 36$ | $|\mathcal{L}| = 54$ |
| `Llama3.1-8B` | 1e-05 | 0.001 | 0.003 | 0.02 | 0.089 | 0.115 | 0.151 | 0.198 |
| `Qwen2.5-7B` | 0 | 2e-05 | 4e-04 | 0.006 | 0.077 | 0.106 | 0.140 | 0.177 |
| `DeepSeek-R1Distill-Qwen-7B` | 0 | 5e-04 | 0.001 | 0.02 | 0.038 | 0.048 | 0.069 | 0.100 |
| `Llama3.3-70B` | 0 | 0 | 0 | 4e-04 | 0.009 | 0.003 | 0.017 | 0.084 |
| `GPT-4o-mini` | 0 | 2e-05 | 7e-04 | 1e-04 | 0 | 0.008 | 0.017 | 0.084 |
| `GPT-4.1-mini` | 0 | 0 | 1e-05 | 1e-04 | 0 | 5e-06 | 5e-05 | 0.016 |

*mode* task involves identifying the most frequent outcome given a specified condition, requiring models to filter and reason over a subset of the distribution. Table 1 reports model accuracy as the joint space size $|\mathcal{L}|$ increases. Although *cond-mode* involves fewer possible outputs, its added conditional reasoning makes it more challenging, leading to slightly lower accuracy than *joint-mode*. While `Llama3.3-70B` and `GPT4.1-mini` remain near-perfect, smaller models degrade as the joint size grows. Overall, although models can handle simpler instances of these tasks using their pretraining knowledge, their accuracy drops noticeably as tasks become more complicated. To mitigate this, we conducted additional experiments using one-shot prompting instead of the default zero-shot setting. The results show a substantial improvement in performance, underscoring the effectiveness of few-shot learning for these tasks. The details of these experiments can be found in Appendix E.

**Maximum Likelihood Estimation:** In this task, models are asked to predict either the probability of all joint outcomes *(joint-MLE)* or the conditional probabilities $P(X_q = x_q \mid X_c = x_c)$ for each $x_q \in \mathcal{L}_q$ *(cond-MLE)*. For evaluation, the TVD is computed for each prompt, and the average is reported over 100 prompts. As shown in Table 2, the *joint-MLE* task is relatively straightforward for all models, with even smaller models achieving acceptable TVD values across all joint sizes. In contrast, the *cond-MLE* task is more challenging, as it requires models to derive conditional probabilities from the joint distribution, leading to consistently higher TVD values and a clear gap between smaller and larger models. To mitigate this challenge, we also ran one-shot prompting experiments, which significantly reduced TVD values, showing that even a simple example can boost performance on probabilistic reasoning tasks. Full details of these experiments are provided in Appendix E.

**Sample Generation:** This task evaluates the generative abilities of LLMs by requiring them to generate $K'$ samples from either the joint distribution (*joint-sampling*) or a conditional distribution (*cond-sampling*). We set $K' := 7 \times |\mathcal{L}|$ to ensure that the generated samples are sufficient to approximate the target distribution. Since it is inherently difficult to reproduce a target empirical distribution with a finite number of samples, we also compute the TVD between $K'$ samples drawn from Python's random library and the true distribution and use it as a baseline to compare the quality of samples generated by the models. As shown in Figure 2, TVD in the *joint-sampling* task increases with $|\mathcal{L}|$. Interestingly, `Llama3.3-70B` and `GPT4.1-mini` achieve TVD values even lower than those from Python's random sampler, which includes sampling noise, suggesting that the underlying distribution of their generated samples is remarkably close to the target. In contrast, all models struggle with the *cond-sampling* task, as it implicitly requires the models to first estimate the conditional probabilities and then generate samples that align with the resulting distribution. Surprisingly, even though no model reaches an acceptable performance, `DeepSeek-R1-Distill-Qwen`, designed for reasoning tasks, outperforms other strong models.

Beyond distributional evaluation, we also examined whether model-generated samples were truly independent. Using the Durbin–Watson statistic and transition matrix entropy (Durbin & Watson, 1951; Shannon, 1948), we found correlations in all models' outputs. In a related experiment, when `GPT-4.1-mini` was prompted to generate one sample per prompt, it strongly favored high-frequency outcomes while rarely producing rare ones. These results suggest that while capable LLMs can generate samples that closely match target empirical probabilities, they struggle to produce independent samples, consistent with the autoregressive nature of LLMs. To further validate our findings, we repeated all six task configurations on a highly-skewed subset of the Mushroom dataset (mus, 1981), with results aligning closely to those from synthetic data (see Appendices F, G).

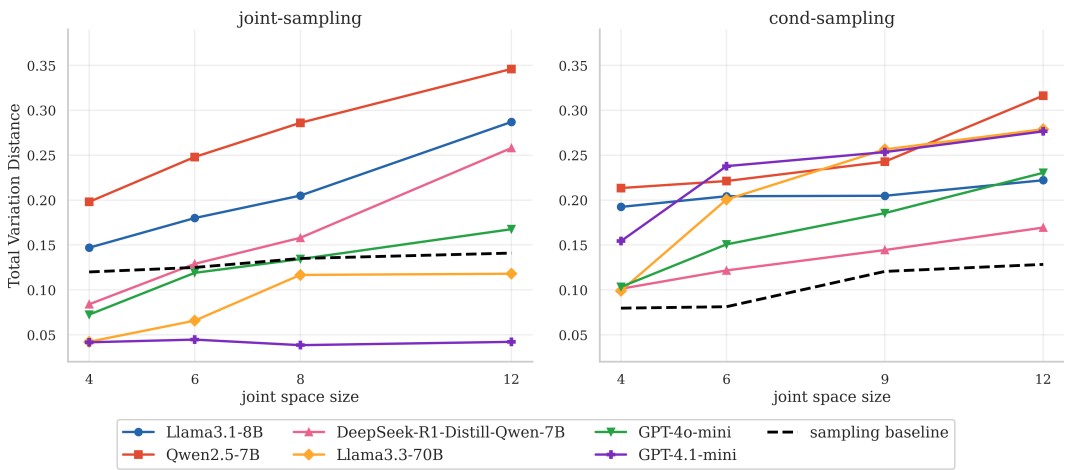

Figure 2: Average TVD across 100 prompts for *joint-sampling* (left) and *cond-sampling* (right) tasks, with models queried to generate $7 \times |\mathcal{L}|$ samples. Here, the sampling baseline is computed as the TVD between $K'$ samples generated by Python's random function and the true distribution.

## 5.4 ROBUSTNESS ANALYSIS RESULTS

This section presents the analysis of two key aspects: the sensitivity of models to variations in outcomes of the random variables, and the effect of providing raw samples in the prompt instead of their empirical frequencies. These analyses shed light on how superficial changes and input formats can significantly affect performance.

### 5.4.1 LABEL SENSITIVITY RESULTS

To investigate the impact of label choice on model performance, a targeted set of experiments is conducted on three models: the two top-performing models, `Llama3.3-70B` and `GPT-4.1-mini`, and a smaller model, `Qwen2.5-7B`. All three are evaluated on the *joint-mode* task, which they handle reliably under standard conditions, using joint space size of $|\mathcal{L}| = 54$ and ten distinct label categories. These categories span a broad range of semantics, including computer science terms, human names, US states, countries, fruits, universities, majors, car brands, neurology terms, and food. By systematically varying the labels used to represent the outcomes of the random variables, we aimed to assess how surface-level changes in label representation influence model behavior.

Figure 3 (right) presents the accuracy of each model across the different label categories. `GPT-4.1-mini` shows remarkable consistency, with minimal variation across labels, indicating strong robustness. `Llama3.3-70B` performs well overall but exhibits moderate sensitivity, with performance varying more noticeably across label sets. In contrast, `Qwen2.5-7B` is highly sensitive to label changes, suffering sharp drops in accuracy depending on the category. These findings suggest that while larger models or those distilled from them can generalize well across prompt variations, smaller models are significantly vulnerable to the way information is presented, even when the underlying task remains unchanged.

### 5.4.2 SAMPLES-IN-CONTEXT RESULTS

Next, we investigate how model performance is affected when raw observed samples are provided in the prompt instead of their empirical frequencies. This setting is evaluated using the two best-performing models, `Llama3.3-70B` and `GPT-4.1-mini`, as smaller models consistently fail under these conditions. We repeat the *joint-mode* task across different joint space sizes, but rather than supplying frequency counts for each outcome, we provide the raw samples directly as mentioned. This forces the models to perform implicit counting, first identifying the frequency of each outcome from the raw data, and then determining the mode based on that.

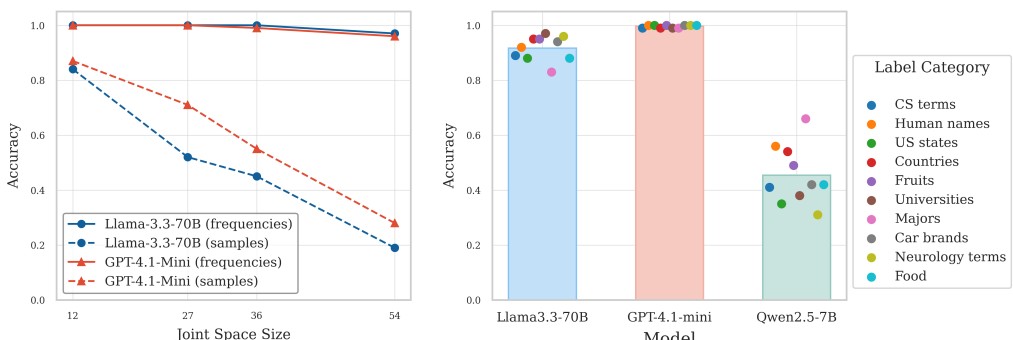

Figure 3: **(Left)** Accuracy comparison on the *joint-mode* task when models are given either raw samples or frequency counts. Accuracy sharply declines with increasing joint space size when raw samples are used, revealing the difficulty of reasoning over longer contexts without explicit frequency information. `GPT-4.1-mini` shows greater robustness than `Llama3.3-70B` in this setting. **(Right)** Accuracy of models across ten different label categories in the *joint-mode* task. `GPT-4.1-mini` remains consistently robust across categories, while `Qwen2.5-7B` exhibits significant sensitivity.

The empirical evaluations reveal a sharp decline in performance. Both models, which had previously achieved near-perfect accuracy with explicit frequencies, struggle significantly in this setting. As the joint space size $|\mathcal{L}|$ increases and the input grows longer, their ability to track counts and identify the correct outcome diminishes rapidly. Figure 3 (left) illustrates this trend, showing an almost linear decrease in accuracy with increasing joint size. Among the two, `GPT-4.1-mini` consistently outperforms `Llama3.3-70B`, indicating greater robustness to the added complexity and input length. Further evidence from Appendix H shows that even when the joint space size is fixed at $|\mathcal{L}| = 12$, increasing the number of observed samples $K$ results in further degradation of accuracy. This suggests that the challenge is not only tied to task complexity but also to the models' limited capacity for reasoning over long sequences.

To explore potential mitigation strategies, we conducted an additional experiment with `GPT-4.1-mini` where the model was given access to a code interpreter. In this setup, the model was encouraged to write Python code to count frequencies before solving the task. This hybrid approach substantially improved performance, nearly matching the results from the frequency-provided setting. The detailed results of these experiments are reported in Appendix I.

## 6 CONCLUSION

This work presents the first comprehensive evaluation of large language models' abilities to understand discrete probability distributions. Through a set of structured tasks, mode identification, maximum likelihood estimation, and sample generation, we reveal clear differences in performance across models. Larger models, or those distilled from them, generally perform well, with sample generation emerging as a surprising strength, where in some cases, generated samples align with the target distribution even more closely than those from a true random sampler. Yet, notable weaknesses remain. Performance consistently declines when tasks involve conditional distributions rather than joint distributions, reflecting the difficulty of conditional reasoning. Models also show sensitivity to superficial changes in the outcomes of the random variables and struggle when asked to reason over long contexts or to infer probabilities directly from raw samples. Taken together, these findings suggest that while pretraining knowledge enables LLMs to handle probabilistic tasks, their performance declines as the complexity of the problems grows. In such cases, providing additional support, such as in-context examples or external tools like a code interpreter, can help boost the performance. Addressing these challenges is a key step toward developing LLMs capable of robust reasoning under uncertainty. Additional discussion of limitations and considerations is provided in Appendix A.

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

## APPENDIX

This section provides additional details, prompts, and results that supplement the main text, beginning with a discussion of limitations and future directions. Section B presents complete prompt examples for each task, while Section C lists the exact frequency distributions used for different joint space sizes. Section D describes the LLM judge prompt template employed to evaluate outputs. Section E explores one-shot prompting, including examples and results, and Section F examines whether samples generated by LLMs are independent. Section G reports evaluations on a real-world dataset to assess generalizability, Section H provides supplemental results on the effect of sample count in context and Section I details experiments where GPT-4.1-mini was given access to a code interpreter.

## A  LIMITATIONS AND FUTURE DIRECTIONS

This study can extend in many directions, each requiring substantial time, effort, and computational resources. To maintain focus and feasibility, we limited our scope to a subset of these directions and aimed to investigate them thoroughly. One key limitation is the scale of the joint distributions used in our experiments. We limit the joint space size at 54 since smaller models struggled with longer contexts, leading to unstable and unreliable performance. Scaling to larger joint spaces would further degrade performance and reduce comparability across model sizes. Another limitation lies in our choice of distribution types. While analyzing both discrete and continuous distributions would provide a more complete picture of LLMs' capabilities, we deliberately focused on discrete distributions. We identified a significant gap in understanding LLMs' reasoning over discrete distributions, and we believe this gap deserves focused attention as a foundational problem that presents distinct challenges from continuous distributions. Finally, we restricted our exploration to a limited subset of distributions due to the substantial effort and time required to evaluate a broader range. Future work is needed to explore and uncover these remaining directions.

# B    TASK PROMPTS

Each task is evaluated with two prompt variants: one for reasoning over joint distributions and one for conditional distributions. We designed three structured tasks to evaluate different aspects of probabilistic reasoning. Below are the representative prompts used for each task.

## B.1    MODE IDENTIFICATION

Below, is an example of a prompt used for *joint-mode* task for $|\mathcal{L}| = 12$:

---

Consider 3 discrete random variables X, Y, Z, where X can take outcomes from A, B, Y can take outcomes from C, D, Z can take outcomes from E, F, G. In 60 independent samples drawn from their joint distribution P(X, Y, Z), the observed frequencies are:

- (A, C, E): 5
- (A, C, F): 7
- (A, C, G): 3
- (A, D, E): 10
- (A, D, F): 2
- (A, D, G): 3
- (B, C, E): 6
- (B, C, F): 8
- (B, C, G): 2
- (B, D, E): 3
- (B, D, F): 3
- (B, D, G): 8

Task: Identify the mode (most probable outcome) of the joint distribution P(X, Y, Z).

Instructions:

1. Do not write any code or pseudocode.
2. Strictly follow the output format below.
3. You may explain your reasoning, but the final answer should be explicitly summarized at the end. You get negative penalty for not following the output format.

Output Format that you should strictly follow:

Mode = (, , )

---

Below is a variation of the above prompt that requires solving the task on a conditional distribution (*cond-mode*).

---

Consider 3 discrete random variables X, Y, Z, where X can take outcomes from A, B, Y can take outcomes from C, D, Z can take outcomes from E, F, G. In 60 independent samples drawn from their joint distribution P(X, Y, Z), the observed frequencies are:

- (A, C, E): 5
- (A, C, F): 7
- (A, C, G): 3
- (A, D, E): 10
- (A, D, F): 2
- (A, D, G): 3
- (B, C, E): 6
- (B, C, F): 8
- (B, C, G): 2
- (B, D, E): 3
- (B, D, F): 3
- (B, D, G): 8

Task: Identify the mode (most probable outcome) of the conditional distribution $P(Z \mid X = A)$.

Instructions:

1. Do not write any code or pseudocode.
2. Strictly follow the output format below.
3. You may explain your reasoning, but the final answer should be explicitly summarized at the end. You get negative penalty for not following the output format.

Output Format that you should strictly follow:

Most probable value of Z given X=A is

---

## B.2 Maximum Likelihood Estimation

The model is asked to estimate the probabilities of the joint distributions using maximum likelihood estimates from observed frequencies.

A prompt sample for *joint-MLE* task for $|\mathcal{L}| = 12$:

---

Consider 3 discrete random variables X, Y, Z, where X can take outcomes from A, B, Y can take outcomes from C, D, Z can take outcomes from E, F, G. In 60 independent samples drawn from their joint distribution P(X, Y, Z), the observed frequencies are:

- (A, C, E): 5
- (A, C, F): 7
- (A, C, G): 3
- (A, D, E): 10
- (A, D, F): 2
- (A, D, G): 3
- (B, C, E): 6
- (B, C, F): 8
- (B, C, G): 2
- (B, D, E): 3
- (B, D, F): 3
- (B, D, G): 8

Task: Predict the maximum likelihood estimation (MLE) of the joint probability distribution P(X, Y, Z) based on these 60 samples.

Instructions:

1. Think step by step and solve this mathematically using probability theory - do not write any code or pseudocodes as you get negative penalty for that.

2. Clearly state the final estimated probabilities for each (X, Y, Z) outcome. Output probabilities should be expressed as float numbers with up to four decimal points.

3. You may explain your reasoning, but the final answer should be explicitly summarized at the end. only your final answer will be graded.

Output Format:

- Final probabilities as float numbers should be listed as:

P(A, C, E) = [value]
P(A, C, F) = [value]
P(A, C, G) = [value]
P(A, D, E) = [value]
...
P(B, D, F) = [value]
P(B, D, G) = [value]

---

Below is a variation of the above prompt that requires solving the task on a conditional distribution (*cond-MLE*).

---

Consider 3 discrete random variables X, Y, Z, where X can take outcomes from A, B, Y can take outcomes from C, D, Z can take outcomes from E, F, G. In 60 independent samples drawn from their joint distribution P(X, Y, Z), the observed frequencies are:

- (A, C, E): 5
- (A, C, F): 7
- (A, C, G): 3
- (A, D, E): 10
- (A, D, F): 2
- (A, D, G): 3
- (B, C, E): 6
- (B, C, F): 8
- (B, C, G): 2
- (B, D, E): 3
- (B, D, F): 3
- (B, D, G): 8

Task: Predict the maximum likelihood estimation (MLE) of the conditional distribution P(X | Z = E).

Instructions:

1. Think step by step and solve this mathematically using probability theory - do not write any code or pseudocodes as you get negative penalty for that.

2. Clearly state the final estimated probabilities as float numbers with up to four decimal points.

3. You may explain your reasoning, but the final answer should be explicitly summarized at the end.

Output Format:

- Final probabilities as float numbers should be listed as:

P(X=A | Z=E) = [value]
P(X=B | Z=E) = [value]

---

## B.3 SAMPLE GENERATION

In this task, the model is asked to generate samples either from the joint distribution or from a conditional distribution, and we compare the empirical distribution of generated samples with the true distribution.

A prompt sample for *joint-sampling* task for $|\mathcal{L}| = 12$:

---

Consider 3 discrete random variables X, Y, Z, where X can take outcomes from A, B, Y can take outcomes from C, D, Z can take outcomes from E, F, G. In 60 independent samples drawn from their joint distribution P(X, Y, Z), the observed frequencies are:

- (A, C, E): 5
- (A, C, F): 7
- (A, C, G): 3
- (A, D, E): 10
- (A, D, F): 2
- (A, D, G): 3
- (B, C, E): 6
- (B, C, F): 8
- (B, C, G): 2
- (B, D, E): 3
- (B, D, F): 3
- (B, D, G): 8

Task: Generate EXACTLY 84 random samples from the joint distribution P(X, Y, Z).

Instructions:

1. OUTPUT MUST BEGIN with ### Output on a new line.
2. List EXACTLY 84 samples numbered 1 to 84.
3. Do not write any code or pseudocode.
4. Strictly follow the output format below.
5. You will be penalized for writing codes or not following the output format.

Output Format that you should strictly follow:

### Output
1. (X, Y, Z)
2. (X, Y, Z)
...
84. (X, Y, Z)

Where X can take outcomes from A, B, Y can take outcomes from C, D, Z can take outcomes from E, F, G.

---

Below is a variation of the above prompt that requires solving the task on a conditional distribution (*cond-sampling*).

---

Consider 3 discrete random variables X, Y, Z, where X can take outcomes from A, B, Y can take outcomes from C, D, Z can take outcomes from E, F, G. In 60 independent samples drawn from their joint distribution P(X, Y, Z), the observed frequencies are:

- (A, C, E): 5
- (A, C, F): 7
- (A, C, G): 3
- (A, D, E): 10
- (A, D, F): 2
- (A, D, G): 3
- (B, C, E): 6
- (B, C, F): 8
- (B, C, G): 2
- (B, D, E): 3
- (B, D, F): 3
- (B, D, G): 8

Task: Generate EXACTLY 84 random samples from the conditional distribution P(Z | X=A).

Instructions:

1. OUTPUT MUST BEGIN with ### Output on a new line.
2. List EXACTLY 84 samples numbered 1 to 84.
3. Do not write any code or pseudocodes.
4. Strictly follow the output format below.
5. You will be penalized for writing codes or not following the output format.

Output Format that you should strictly follow:

### Output
1. Z
2. Z
...
84. Z
Where Z can take outcomes from E, G, F.

---

## C    DISTRIBUTIONS USED FOR EACH JOINT SPACE SIZE

Distribution used for $|\mathcal{L}| = 4$

("A", "C"): 6, ("A", "D"): 4, ("B", "C"): 8, ("B", "D"): 2

Distribution used for $|\mathcal{L}| = 6$

("A", "C"): 7, ("A", "D"): 2, ("A", "E"): 4, ("B", "C"): 9, ("B", "D"): 2, ("B", "E"): 6

Distribution used for $|\mathcal{L}| = 8$

("A", "C", "E"): 7, ("A", "C", "F"): 4, ("A", "D", "E"): 9, ("A", "D", "F"): 2,
("B", "C", "E"): 4, ("B", "C", "F"): 3, ("B", "D", "E"): 6, ("B", "D", "F"): 5

Distribution used for $|\mathcal{L}| = 9$

("A", "D"): 6, ("A" , "E"): 3, ("A", "F"): 4, ("B", "D"): 9, ("B", "E"): 2, ("B", "F"): 5, ("C", "D"):
4, ("C", "E"): 3, ("C", "F"): 4

Distribution used for $|\mathcal{L}| = 12$

("A", "C", "E"): 8, ("A", "C", "F"): 2, ("A", "C", "G"): 7, ("A", "D", "E"): 3,
("A", "D", "F"): 3, ("A", "D", "G"): 3, ("B", "C", "E"): 6, ("B", "C", "F"): 10,
("B", "C", "G"): 2, ("B", "D", "E"): 8, ("B", "D", "F"): 5, ("B", "D", "G"): 3,

Distribution used for $|\mathcal{L}| = 27$

("A", "D", "G"): 8, ("A", "D", "H"): 4, ("A", "D", "I"): 2, ("A", "E", "G"): 3, ("A", "E", "H"): 4,
("A", "E", "I"): 5, ("A", "F", "G"): 6, ("A", "F", "H"): 3, ("A", "F", "I"): 6, ("B", "D", "G"): 10,
("B", "D", "H"): 6, ("B", "D", "I"): 1, ("B", "E", "G"): 7, ("B", "E", "H"): 2, ("B", "E", "I"): 4,
("B", "F", "G"): 6, ("B", "F", "H"): 6, ("B", "F", "I"): 3, ("C", "D", "G"): 4, ("C", "D", "H"): 5,
("C", "D", "I"): 6, ("C", "E", "G"): 7, ("C", "E", "H"): 4, ("C", "E", "I"): 2, ("C", "F", "G"): 3,
("C", "F", "H"): 8, ("C", "F", "I"): 5 ,

Distribution used for $|\mathcal{L}| = 36$

("A", "E", "H"): 6, ("A", "E", "I"): 8, ("A", "E", "J"): 13, ("A", "F", "H"): 9, ("A", "F", "I"): 1,
("A", "F", "J"): 2,("A", "G", "H"): 11, ("A", "G", "I"): 2, ("A", "G", "J"): 1, ("B", "E", "H"): 15,
("B", "E", "I"): 4, ("B", "E", "J"): 8, ("B", "F", "H"): 1, ("B", "F", "I"): 5, ("B", "F", "J"): 5,
("B", "G", "H"): 1, ("B", "G", "I"): 7, ("B", "G", "J"): 2, ("C", "E", "H"): 4, ("C", "E", "I"): 5,
("C", "E", "J"): 12, ("C", "F", "H"): 3, ("C", "F", "I"): 2, ("C", "F", "J"): 2, ("C", "G", "H"): 4,
("C", "G", "I"): 8, ("C", "G", "J"): 12, ("D", "E", "H"): 2, ("D", "E", "I"): 4, ("D", "E", "J"): 2,
("D", "F", "H"): 1, ("D", "F", "I"): 2, ("D", "F", "J"): 2, ("D", "G", "H"): 9, ("D", "G", "I"): 2,
("D", "G", "J"): 3

Distribution used for $|\mathcal{L}| = 54$

("A", "D", "F", "I"): 1, ("A", "D", "F", "J"): 5, ("A", "D", "F", "K"): 9, ("A", "D", "G", "I"): 1,
("A", "D", "G", "J"): 10, ("A", "D", "G", "K"): 2, ("A", "D", "H", "I"): 3, ("A", "D", "H", "J"): 2,
("A", "D", "H", "K"): 5, ("A", "E", "F", "I"): 1, ("A", "E", "F", "J"): 1, ("A", "E", "F", "K"): 5,
("A", "E", "G", "I"): 5, ("A", "E", "G", "J"): 12, ("A", "E", "G", "K"): 2, ("A", "E", "H", "I"): 6,
("A", "E", "H", "J"): 4, ("A", "E", "H", "K"): 2, ("B", "D", "F", "I"): 10, ("B", "D", "F", "J"): 2,
("B", "D", "F", "K"): 1, ("B", "D", "G", "I"): 7, ("B", "D", "G", "J"): 2, ("B", "D", "G", "K"): 3,
("B", "D", "H", "I"): 8, ("B", "D", "H", "J"): 2, ("B", "D", "H", "K"): 7, ("B", "E", "F", "I"): 2,
("B", "E", "F", "J"): 2, ("B", "E", "F", "K"): 3, ("B", "E", "G", "I"): 4, ("B", "E", "G", "J"): 5,
("B", "E", "G", "K"): 13, ("B", "E", "H", "I"): 2, ("B", "E", "H", "J"): 4, ("B", "E", "H", "K"): 12,
("C", "D", "F", "I"): 12, ("C", "D", "F", "J"): 1, ("C", "D", "F", "K"): 3, ("C", "D", "G", "I"): 1,

("C", "D", "G", "J"): 15, ("C", "D", "G", "K"): 9, ("C", "D", "H", "I"): 11, ("C", "D", "H", "J"): 1, ("C", "D", "H", "K"): 4, ("C", "E", "F", "I"): 1, ("C", "E", "F", "J"): 5, ("C", "E", "F", "K"): 3, ("C", "E", "G", "I"): 11, ("C", "E", "G", "J"): 13, ("C", "E", "G", "K"): 3, ("C", "E", "H", "I"): 7, ("C", "E", "H", "J"): 4, ("C", "E", "H", "K"): 1

## D    LLM JUDGE PROMPT

We used the LLM judge only for the mode identification task, where model responses were often difficult to evaluate using rule-based methods. Many responses implied an answer but did not state it clearly or follow a consistent format, making extraction with regular expressions unreliable. For the other two tasks—MLE and sampling—models typically followed the expected output format, or when they did not, their responses clearly lacked an answer, making them easy to discard.

To ensure the judge's reliability, we compared its decisions with human judgments on 100 randomly selected responses. The judge matched human decisions in 99 cases, indicating its reliable judgments.

To automate evaluation, the judge model receives the task, expected output, and model response, and decides whether the response is correct.

Below, is the **system prompt** used for our judge LLM, `Llama3.3-70B`

> You are an expert evaluator assessing the correctness of responses. Your task is to judge whether a response is correct based on an expected answer. If the response correctly conveys the intended meaning, mark it Correct. Otherwise, mark it Incorrect.

The template of the **user prompt** that is provided to the judge model is given below:

> Given the following question, expected answer, and response, judge whether the response is correct. Clearly state your judgment as "Judgment: Correct." or "Judgment: Incorrect.".
>
> Question: [question]
>
> Expected Answer: [expected]
>
> Response to Evaluate: [response]
>
> Judgment (Correct/Incorrect):

Table 3: Performance comparison of different models in zero-shot vs one-shot settings across four tasks with $|\mathcal{L}| = 54$. Mode identification tasks are evaluated using accuracy, while MLE tasks are evaluated using total variation distance (TVD).

| Model | *joint-mode* | | *cond-mode* | | *joint-MLE* | | *cond-MLE* | |
|---|---|---|---|---|---|---|---|---|
| | zero-shot | one-shot | zero-shot | one-shot | zero-shot | one-shot | zero-shot | one-shot |
| Llama3.1-8B | 0.21 | **0.88** | 0.62 | **0.93** | 0.02 | **0** | 0.198 | **0.082** |
| Qwen2.5-7B | 0.65 | **0.98** | 0.60 | **0.87** | 0.006 | **6e-05** | 0.177 | **0.041** |
| DeepSeek-R1Distill-Qwen-7B | 0.86 | **0.9** | 0.83 | **0.95** | 0.02 | **0.001** | 0.100 | **0.09** |
| Llama3.3-70B | 0.97 | **1.0** | 0.95 | **1.0** | 4e-04 | **0** | 0.084 | **0.004** |
| GPT-4o-mini | 0.80 | **1.0** | 0.88 | **1.0** | 1e-04 | **0** | 0.084 | **0.003** |
| GPT-4.1-mini | 0.96 | **1.0** | 1.0 | **1.0** | 1e-04 | **0** | 0.016 | **3e-04** |

# E ONE-SHOT PROMPTING

As illustrated in Tables 1 and 2, models tend to struggle on the more complex tasks when relying solely on their pretraining knowledge. To address this, we designed a set of experiments in which each prompt was augmented with a single, simpler example of the task. Specifically, we included an example with $|\mathcal{L}| = 12$ along with its step-by-step solution, followed by the target, more challenging task with $|\mathcal{L}| = 54$, and asked the model to solve it.

As shown in Table 3, this one-shot prompting strategy substantially improved the models' accuracy, highlighting the impact of providing minimal guidance on solving the harder problems. Interestingly, we observed that the performance of smaller models with just one in-context example closely matched the zero-shot performance of much larger models, underscoring the strength of in-context learning as a way to compensate for smaller model size.

An example of the one-shot prompt used in our experiments is provided below.

---

**Task Definition:** Consider the joint probability distribution of a set of discrete random variables. You will be provided with the frequencies of each outcome of the joint distribution in a set of independently drawn samples. Your task is to identify the mode (the most probable outcome) of a specific conditional distribution.

— EXAMPLE with Solution—

Consider discrete random variables X, Y, Z with joint distribution P(X, Y, Z) where X can take outcomes from A, B, Y can take outcomes from C, D, Z can take outcomes from E, F, G. In 60 independent samples drawn from the joint distribution P(X, Y, Z), the observed frequencies are:
- (A, C, E): 5
- (A, C, F): 7
- (A, C, G): 3
- (A, D, E): 10
- (A, D, F): 2
- (A, D, G): 3
- (B, C, E): 6
- (B, C, F): 8
- (B, C, G): 2
- (B, D, E): 3
- (B, D, F): 3
- (B, D, G): 8

Example Task (solved): Identify the mode (most probable outcome) of the conditional

---

distribution P(Z | X = A).

Step-by-step solution:

1. Extract rows where X = A:
(A,C,E):5, (A,C,F):7, (A,C,G):3, (A,D,E):10, (A,D,F):2, (A,D,G):3.

2. Aggregate counts by Z among those rows:
- Count(Z = E | X = A) = 5 + 10 = 15.
- Count(Z = F | X = A) = 7 + 2 = 9.
- Count(Z = G | X = A) = 3 + 3 = 6.

3. Total samples with X = A = 15 + 9 + 6 = 30.

4. Compute empirical conditional probabilities.
- P(Z = E | X = A) = 15 / 30 = 0.5
- P(Z = F | X = A) = 9 / 30 = 0.3
- P(Z = G | X = A) = 6 / 30 = 0.2

5. Identify the mode: Z = E (highest probability 0.5).

Final Answer:
Most probable value of Z given X=A is E
— END EXAMPLE —

Main Task:
Consider 4 discrete random variables X, Y, Z, T, where X can take outcomes from A, B, C, Y can take outcomes from D, E, Z can take outcomes from F, G, H, T can take outcomes from I, J, K. In 270 independent samples drawn from their joint distribution P(X, Y, Z, T), the observed frequencies are:

- (A, D, F, I): 6
- (A, D, F, J): 3
...
- (C, E, H, I): 4
- (C, E, H, J): 2
- (C, E, H, K): 1

Main Task (to solve): Identify the mode (most probable outcome) of the conditional distribution P(X | Y = E).

Instructions:
1. Do not write any code or pseudocodes.
2. Strictly follow the output format below.
3. You may explain your reasoning, but the final answer should be explicitly summarized at the end. You get negative penalty for not following the output format.

Output Format that you should strictly follow:
Most probable value of X given Y=E is

Table 4: Independence analysis of generated samples by each model. Reported values include the Durbin–Watson (DW) statistic and transition matrix entropy. Samples are classified as non-independent if DW < 1.5 or > 2.5, or if the entropy ratio < 0.8.

| Model | (DW) statistic | entropy-ratio | Is independent? |
|---|---|---|---|
| Llama-3.1-8B-Instruct | 0.118 | 0.245 | No |
| Qwen2.5-7B-Instruct-1M | 0.002 | 0.209 | No |
| DeepSeek-R1-Distill-Qwen-7B | 0.002 | 0.195 | No |
| Llama3.3-70B | 0.153 | 0.323 | No |
| GPT-4o-mini | 0.636 | 0.608 | No |
| GPT-4.1-mini | 0.149 | 0.336 | No |

## F  INDEPENDENCE OF GENERATED SAMPLES

In our observations, LLM-generated samples are not fully independent and often follow ordering patterns similar to those in the input prompt. For example, as illustrated below, DeepSeek-R1-Distill-Qwen explicitly acknowledges the need to shuffle the samples. However, due to the lack of access to a code interpreter or true randomness, it prioritizes matching target frequencies over enforcing independence:

*"So, the final step is to create a list of 84 triplets, where each triplet is repeated according to its count, and then the list is shuffled to randomize the order. But since I can't perform the shuffling here, I'll have to present the triplet counts and note that the actual output would be a shuffled list"*

To investigate this phenomenon more systematically, we conducted additional analyses using two statistical methods: (1) the Durbin–Watson (DW) statistic (Durbin & Watson, 1951) to detect autocorrelation, and (2) transition matrix entropy (Shannon, 1948) to evaluate diversity in transition patterns. Samples are classified as non-independent based on the aggregated results of these two measures. Table 4 reports the corresponding statistics for samples generated by each model.

Table 5: Frequencies of joint distribution outcomes for the selected subset of the Mushroom dataset. Each column corresponds to one outcome of the joint distribution, along with its frequency in 80 samples. The distribution is highly skewed, with several outcomes having zero frequency.

| Mushroom Feature | Labels | | | | | | | | | | | | | | | |
|---|---|---|---|---|---|---|---|---|---|---|---|---|---|---|---|---|
| gill-attachment | f | f | f | f | f | f | f | f | a | a | a | a | a | a | a | a |
| gill-size | b | b | b | b | n | n | n | n | b | b | b | b | n | n | n | n |
| gill-color | k | g | p | o | k | g | p | o | k | g | p | o | k | g | p | o |
| **Frequency** | 13 | 17 | 34 | 0 | 2 | 0 | 9 | 0 | 0 | 0 | 0 | 0 | 0 | 0 | 0 | 0 |

Table 6: Performance of different models across all six task configurations on the Mushroom dataset. Mode identification tasks are evaluated using accuracy, while MLE and sampling tasks are evaluated using total variation distance (TVD).

| Model | joint-mode | cond-mode | joint-MLE | cond-MLE | joint-samp | cond-samp |
|---|---|---|---|---|---|---|
| Llama-3.1-8B-Instruct | 80 | 78 | 0.011 | 0.111 | 0.465 | 0.414 |
| Qwen2.5-7B-Instruct-1M | 95 | 75 | 0.003 | 0.131 | 0.544 | 0.441 |
| DeepSeek-R1-Distill-Qwen-7B | 96 | 96 | 0.002 | 0.100 | 0.237 | 0.173 |
| Llama3.3-70B | 100 | 100 | 0.0 | 0.005 | 0.111 | 0.089 |
| GPT-4o-mini | 100 | 91 | 0.0 | 0.028 | 0.287 | 0.352 |
| GPT-4.1-mini | 100 | 100 | 0.0 | 0.0 | 0.193 | 0.218 |

## G  EVALUATION ON REAL-WORLD DATA

To validate our findings on real-world data, we conducted experiments on the Mushroom dataset (mus, 1981), focusing on three categorical features and a subset of 80 samples. The resulting joint distribution is highly skewed, with a large proportion of zero-frequency outcomes, making the task particularly challenging. Table 5 summarizes the observed outcomes and their frequencies in this subset. The experimental results, detailed in Table 6, show that larger models, as well as distilled variants, consistently outperform smaller models across all task configurations.

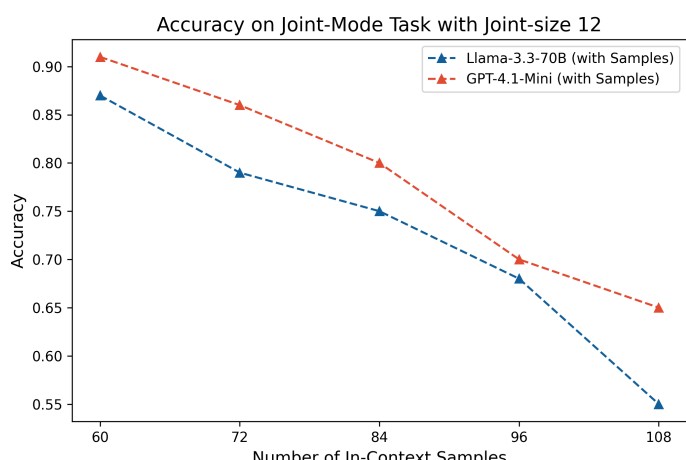

Figure 4: Accuracy on *joint-mode* task with fixed joint space size $|\mathcal{L}| = 12$ as the number of observed in-context samples increases.

## H EFFECT OF SAMPLE COUNT IN CONTEXT

To investigate the impact of input length independently from task complexity, we fixed the joint space size at $|\mathcal{L}| = 12$ and varied the number of observed samples $K$ provided in the prompt. As shown in Figure 4, increasing the number of in-context samples leads to a clear drop in accuracy for both models. This degradation suggests that the challenge stems not just from the probabilistic reasoning task itself, but also from the models' limited capacity to process and reason over longer contexts. This highlights a core limitation in current LLMs and points to context scaling as a key direction for future improvement.

Table 7: Accuracy of GPT-4.1-mini on the *joint-mode* task under three configurations across different joint sizes. Results show that accuracy in the samples-in-context setting with access to code interpreter closely matches the frequencies-provided case.

| Setting | $|\mathcal{L}| = 12$ | $|\mathcal{L}| = 27$ | $|\mathcal{L}| = 36$ | $|\mathcal{L}| = 54$ |
|---|---|---|---|---|
| Frequencies in context | 1.0 | 1.0 | 0.99 | 0.96 |
| Raw samples in context | 0.87 | 0.71 | 0.55 | 0.28 |
| Raw samples + code interpreter | 1.0 | 1.0 | 0.99 | 0.97 |

## I  ACCESS TO CODE INTERPRETER

In the frequency-based setting, both Llama3.3-70B and GPT-4.1-mini achieved near-perfect accuracy, but their performance degraded sharply when asked to infer frequencies directly from raw samples due to limited counting abilities. To explore mitigation strategies, we conducted an additional set of experiments with GPT-4.1-mini augmented with access to a code interpreter. In this setup, prompts encouraged the model to write Python code to count outcome frequencies before solving the *joint-mode* task. This hybrid approach led to a substantial performance gain. As shown in Table 7, accuracy levels were restored to nearly those observed in the frequency-provided condition, demonstrating that the main limitation lies in the models' ability to perform reliable counting over long contexts rather than in reasoning about the task itself.

