# OpenReview forum: "Reasoning Under Uncertainty: Exploring Probabilistic Reasoning Capabilities of LLMs"
_ICLR.cc/2026/Conference — ICLR 2026 Conference Withdrawn Submission_

### Official Review · Reviewer_t8CH · 2025-10-29

**Soundness:** 3
**Presentation:** 4
**Contribution:** 3
**Rating:** 6
**Confidence:** 4

**Summary:**

This paper presents a comprehensive study of how LLMs reason over explicit discrete probability distributions. The authors construct a new framework featuring three carefully designed tasks—mode identification, MLE and sample generation. These tasks are used to evaluate models on both joint and conditional distributions, probing a range of skills from frequency analysis to generative behavior.

**Strengths:**

This paper addresses an important question regarding the fundamental capabilities of LLMs. The authors have carefully designed three evaluation tasks and a robust experimental setup, which yield solid results and insightful conclusions. The paper is well-written and easy to follow.

**Weaknesses:**

1. The evaluation framework is highly dependent on specific prompt engineering. Optimizing the prompt format for each task is great, but performance may be brittle to superficial changes in the prompt, limiting the generality of the conclusions.

2. The paper highlights the performance gap between larger and smaller models as a key finding. This conclusion about scaling trends, while true, is largely expected. This finding is less novel than the paper's other insight on mechanism of failure.

**Questions:**

In general a conditional task is harder than joint one, but why in table 1 does llama3.1-8b have better performance on the conditional mode task?

How would reasoning benefit this task? Can you give more analysis between qwen 7b and deepseek distilled qwen 7b (e.g. answer pattern between instructed model and reasoning model)?

In the cond-MLE task, is there evidence that models are performing Bayesian inference or are they appearing to directly give answers?

Given that joint-mode with provided frequencies is just a max-finding task, how much does it truly test "probabilistic reasoning" as opposed to simple numerical comparison?

Curious that even with joint size=54 it is still within model context length? Why does the model performance drop severely?

The paper measures the accuracy (TVD) of probability estimates, but are these outputs also calibrated?

---

### Official Review · Reviewer_S4vc · 2025-10-31

**Soundness:** 2
**Presentation:** 3
**Contribution:** 1
**Rating:** 2
**Confidence:** 4

**Summary:**

This paper presents a benchmark for evaluating the probabilistic reasoning capabilities of LLMs. The authors construct a small dataset of synthetic probability distributions and evaluate a set of models (GPT-4.1-mini, Llama3.3-70B, DeepSeek, etc.) on tasks such as mode identification, maximum likelihood estimation, and sample generation. The paper empirically finds that larger models perform better on these tasks, conditional reasoning remains challenging, and that models sometimes generate samples that align very well with the target distribution.

**Strengths:**

1. **Clear presentation.**  The paper is cleanly structured, and the dataset and prompts are provided in a reproducible way. From an engineering perspective, the evaluation pipeline is easy to follow.
2. **A relevant topic for LLM evaluation.** Evaluating the model's capability on the propose tasks is a meaningful research direction.

**Weaknesses:**

1. **Trivial setup and predictable conclusions.** The experiments consist solely of simple discrete probability tables and symbolic counting tasks. Unsurprisingly, larger models or those distilled from them perform better. This is a known and thoroughly established pattern across virtually every reasoning benchmark. The results provide no new insights into model architecture, training, or reasoning mechanisms.

2. **The constrain that the model should not write code is not reasonable.** The authors explicitly forbid models from writing code to finish the task. I do not see why this is reasonable. Given that probabilistic reasoning is inherently computational, this restriction is both arbitrary and non-reasonable.

3. **No methodological innovation.** The paper introduces no new task formulation, reasoning framework, or model adaptation. It is essentially an collection of toy experiments with well-known outcomes. The dataset is too narrow to generalize beyond trivial counting and sampling tasks.

4. **Overinterpretation of obvious results.** Many statements such as *“larger models perform better”* or *“conditional reasoning is harder”* are treated as novel empirical findings rather than baseline expectations. However, these findings should be commonsense instead of the contribution.

5. **Misleading usage of "uncertainty".** Despite the title and repeated mention of “reasoning under uncertainty,” the paper does not actually address uncertainty in epistemic or aleatoric. There is no quantification or calibration of uncertainty within model predictions. The “uncertainty” here merely refers to the existence of a probabilistic dataset, not to any model-level reasoning about uncertainty. This is a serious conceptual mismatch between the title and the actual content.

**Questions:**

Please refer to the weakness section

---

### Official Review · Reviewer_6kdy · 2025-11-01

**Soundness:** 2
**Presentation:** 3
**Contribution:** 3
**Rating:** 4
**Confidence:** 3

**Summary:**

This paper presents a study of the reasoning capabilities of LLMs over explicit discrete probability distributions, focusing on three core tasks: mode identification, maximum likelihood estimation, and sample generation. The authors report a clear performance gap between smaller and larger models, as well as strong sensitivity to the notation used in the prompt to represent probabilistic outcomes. The results highlight ll with counting and long-context reasoning.

**Strengths:**

The paper tackles an important and underexplored question, systematically evaluating how LLMs reason over explicit probability distributions.

The three task setup is intuitive and clearly presented.

**Weaknesses:**

The experimental design does not appear sufficiently robust to serve as a comprehensive evaluation of LLM reasoning capabilities.
* The paper does not sufficiently motivate why these particular tasks were selected as the primary testbed for evaluating probabilistic reasoning.
* The range of model sizes is limited; larger models and reasoning models should be included for a more complete analysis.
* The paper does not clearly specify the dataset size used for each task. “generating ten prompts for each assignment, resulting in 100 prompts per task” is ambiguous ( line 307).
* The set of baselines is limited. The authors should explore additional prompting strategies such as explicit chain-of-thought, self-consistency, instruction-first-then-observerd-samples prompting, or few-shot (one-shot may be insufficient) prompting.
* There is no qualitative analysis of model errors. It would be helpful to separate and quantify the proportions of failures, e.g.,  due to frequency counting errors or reasoning errors or calculation errors under different task scenarios to better understand the underlying weaknesses.

**Questions:**

1. Could you clarify the total number of tokens in the input for Appendix H  in the setup with 108 in-context samples? It is not immediately clear how this qualifies as a long-context setting.

2. Missing citations about LLM probabilistic reasoning: https://aclanthology.org/2023.acl-long.68.pdf, https://openreview.net/pdf?id=fAAaT826Vv

---

### Official Review · Reviewer_upbq · 2025-11-08

**Soundness:** 3
**Presentation:** 3
**Contribution:** 2
**Rating:** 2
**Confidence:** 5

**Summary:**

Summary
The paper presents a benchmark for probabilistic reasoning over discrete distributions using three tasks, mode identification, MLE, and sampling, under joint and conditional settings. It evaluates several modern LLMs and testing settings such as providing frequencies and the samples or raw data.


3/10 clear reject
confidence 5/5

**Strengths:**

Strengths
Comprehensive evaluation with Clear prompts, metrics , and ablations and multiple LLMs.

**Weaknesses:**

Weaknesses / Concerns
The main issue the reviewer has with this paper is the contributions. The paper is trying to assess LLMs in multiple facets, and in doing so, it doesn't dive deep into any of them, leading to generic conclusions such as : the bigger models perform better, in-context examples help with LLMs, and more unnecessary tokens hurt the model performance. These contributions of the paper can apply to any task and are very generic.
Synthetic data: while synthetic data is useful when a paper is pioneering or there is no real data, it’s not the case here. For example, Paruchuri, et al 2024 which is cited in the paper, uses real-world data for sampling.

Minor concerns / weakness:

Section 4 is redundant and discusses results before the experiments section and repeats them in the experiments section again.
The reasoning version of DeepSeek is used without any explanation as to why, while other models are not reasoning.
Suggestions:
The paper should focus on one of the issues mentioned in the paper, like Paruchuri, et al 2024 and provide solutions for the shortcomings of the LLMs not just point out obvious general weaknesses of LLMs that are not even unique to probabilistic reasoning.

**Questions:**

See above.

---

### Note · Authors · 2025-12-03

**Comment:**

After careful consideration, we have decided to withdraw our submission. We respect the review process and appreciate the time reviewers invested in evaluating our work. However, we believe several reviews contain misunderstandings or inaccurate assessments that significantly understate the contributions of our paper. Some critiques suggest that parts of the work may not have been read thoroughly, and several comments do not accurately reflect the content or intent of our study.

Contrary to some impressions, the goal of evaluation studies like ours is to provide empirical insights into the capabilities and limitations of current LLMs, whether by confirming expected behaviors, identifying overlooked weaknesses, or revealing unexpected patterns. Beyond validating some predictable conclusions, our work offers new insights into LLMs’ difficulty with counting (especially in long-context settings), their substantial sensitivity to probabilistic labels, and their unexpectedly strong sampling behavior, where generated samples closely match target distributions. We also analyzed sample independence and uncovered recurring ordering patterns, showing that LLM-generated samples are not independent. We also evaluated two improvement strategies. First, we showed that providing a single simple example (one-shot setting) yields substantial performance gains across all models—even enabling an 8B model to match the zero-shot performance of a 70B model. Second, we assessed the effectiveness of providing a code interpreter, which resulted in significant improvements in additional experiments.

Below are a few examples of reviewer comments that we found unconvincing or inconsistent with the content of our paper:

"The paper does not clearly specify the dataset size used for each task.": these details are explicitly provided in Sections 5.2 and 5.3.
"The range of model sizes is limited.": our experiments include six LLMs ranging from 7B to 70B parameters, including reasoning models.
"The experiments consist solely of simple discrete probability tables and symbolic counting tasks.": our main experiments include joint distributions with sizes ranging from 12 to 54. Counting tasks were included only as an additional analysis.
"The reasoning version of DeepSeek is used without any explanation as to why, while other models are not reasoning": our model selection follows common practice in LLM evaluation work, where the goal is to test widely used and representative models.
"while synthetic data is useful when a paper is pioneering or there is no real data, it’s not the case here.": the claim that synthetic data is inappropriate for our setting is not scientifically supported and contradicts many high-impact evaluation papers that are based on synthetic data.
"The results provide no new insights into model architecture, training, or reasoning mechanisms": architecture and training analysis are outside the scope of evaluation-focused papers, whose purpose is to characterize behavioral strengths and weaknesses.

Given the situation, we believe a rebuttal would not meaningfully change the outcome. Therefore, we have decided to withdraw the paper.

**Withdrawal Confirmation:**

I have read and agree with the venue's withdrawal policy on behalf of myself and my co-authors.